# Effect of Nitrogen Concentration on the Biosynthesis of Citric Acid, Protein, and Lipids in the Yeast *Yarrowia lipolytica*

**DOI:** 10.3390/biom12101421

**Published:** 2022-10-04

**Authors:** Svetlana V. Kamzolova, Julia N. Lunina, Vladimir A. Samoilenko, Igor G. Morgunov

**Affiliations:** G.K. Skryabin Institute of Biochemistry and Physiology of Microorganisms, Pushchino Center for Biological Research of the Russian Academy of Sciences, Prospekt Nauki 5, Moscow Region, 142290 Pushchino, Russia

**Keywords:** microbial synthesis, yeast biochemistry, *Yarrowia lipolytica*, citric acid (CA) production, glucose, nitrogen limitation, biomass production, yeast protein, yeast lipids, fermentation

## Abstract

*Yarrowia lipolytica* yeast is well known to be able to synthesize citric acid (CA) in large amounts. This study deals with CA biosynthesis, the production of biomass, as well as the accumulation and composition of proteins and lipids in *Y. lipolytica* VKM Y-2373 grown in media with glucose at different concentrations of ammonium sulfate (from 2 to 10 g/L). It was found that these concentrations of nitrogen source are limiting for the growth of *Y. lipolytica* and that nitrogen deficiency is the main cause of CA excretion. At the high concentration of (NH_4_)_2_SO_4_ (10 g/L), the accumulation of cell biomass, biomass yield (Y_X/S_), and protein concentration was higher than in the medium with 2 g/L ammonium sulfate by 4.3 times, 143%, and 5.1 times, respectively. CA was accumulated in meaningful quantities only in media containing 3–10 g/L (NH_4_)_2_SO_4_ with the maximum concentration of CA (99.9 g/L) at 4 g/L ammonium sulfate. Also of interest is the technological mode with 6 g/L (NH_4_)_2_SO_4_, which is characterized by high productivity (1.11 g/L × h). It should be noted that biomass contains large amounts of essential amino acids and unsaturated fatty acids and can be used in food biotechnologies and agriculture.

## 1. Introduction

There is increasing interest in citric acid (CA) and its salts in a variety of industries. CA is widely used as an acidifier, flavor enhancer, and buffering agent in the production of various beverages and syrups, as well as for optimal gelation in the production of jams and jellies. CA and sodium citrate are used for casein stabilization in the production of cream, as an emulsifying agent in the cheese industry, and as an improver of the organoleptic properties of dairy products. Sodium citrate inhibits the enzymatic oxidative reactions spoiling frozen fruits and vegetables [1]. CA and sodium citrate are multifunctional pharmaceutical excipients [2] and are used as copolymers in the production of gold nanoparticles [3]. Sodium citrate is also used in the production of gypsum for the stabilization of hydrogen peroxide and vinyl alcohol polymers [4]. Sodium citrate successively replaces tripolyphosphates, especially in liquid laundry detergents. However, in doing so, it is well biodegradable to water and carbon dioxide both in the wastewater treatment plants and open reservoirs [5]. Due to these useful properties of sodium citrate, many countries have banned the production of phosphate-containing detergents.

CA is usually produced with the aid of the opportunistic fungus *Aspergillus niger* cultivated on molasses in the surface or submerged mode. This technology has significant disadvantages because the composition of molasses is unstable. In addition, the isolation of CA produced by this method is accompanied by the formation of great amounts of gypsum, which makes the process ecologically unfavorable [6]. Sodium citrate is commonly produced from CA by its neutralization with NaOH and the isolation of the salt by concentration and crystallization.

Taking all this into account, researchers pay attention to alternative methods of production of CA and its salts. In particular, it is well established that the yeast *Yarrowia lipolytica* grown under nitrogen deficiency and carbon excess is able to produce CA in great amounts from various substrates, such as glucose [7,8,9,10,11,12,13,14,15,16,17], glycerol [18,19,20,21], sucrose [22,23], xylose [24], the mixture of glucose and fructose [25], ethanol [26], vegetable oils [27,28], and so on. The cultivation of *Y. lipolytica* is much easier than that of *A. niger* because the yeast is characterized by a high growth rate and resistance to salts, and various metal ions, as well as to wide ranges of pH and cultivation temperatures [17,29,30,31]. Within this technology, CA and its salts are isolated directly from the culture liquid filtrate [18]. This approach minimizes liquid and solid wastes and eliminates many stages of product isolation and purification. It should be noted that the waste biomass of *Y. lipolytica* remaining after CA isolation is similar to vegetable oils in the fatty acid composition and can be used in the food industry and the production of biodiesel [17]. In general, the yeast *Y. lipolytica* and various products manufactured with its aid are recognized as safe (GRAS) and can be used in the food industry and medicine [31].

There is still no large-scale production of CA yet using the yeast *Y. lipolytica*. The reason is the lack of fundamental knowledge about the mechanism of CA accumulation in the producing yeast culture. Little is known even about the main factor of CA oversynthesis, namely, growth limitation by the nitrogen source. Therefore, conducting research in this direction is of great interest.

The aim of this work was to study the effect of different limiting concentrations of nitrogen on CA biosynthesis, the production of biomass, as well as the accumulation and composition of proteins and lipids in *Y. lipolytica* VKM Y-2373 grown in media with glucose.

## 2. Materials and Methods

### 2.1. Microorganism

Experiments were carried out with the yeast strain *Y. lipolytica* VKM Y-2373 accumulating significant amounts of CA in media with glucose [14].

### 2.2. Medium and Cultivation Conditions

The strain *Y. lipolytica* VKM Y-2373 was grown in the medium containing (g/L): (NH_4_)_2_SO_4_ in different concentrations (see details in the text); MgSO_4_·7H_2_O, 1.4; Ca(NO_3_)_2_·4 H_2_O, 0.8; NaCl, 0.5; KH_2_PO_4_, 2.0; K_2_HPO_4_, 0.2; and microelement solution according to Burkholder [32]. The medium was additionally supplemented with Zn^2+^, Mn^2+^, Cu^2+^ and Fe^2+^ ions in concentrations of 1.2, 0.3, 0.3 and 0.6 mg/L, respectively. Thiamine (200 µg/L) and biotin (20 µg/L) were used as vitamins. Glucose was added in the portions by 20 g/L as it was consumed from the medium. The total amount of glucose in the fermenter for the whole process consisted of 720 g for 2 g/L (NH_4_)_2_SO_4_, 900 g for 3 and 4 g/L (NH_4_)_2_SO_4,_ and 1320 g for 6 and 10 g/L (NH_4_)_2_SO_4._

Cultivation was performed in a 10 L ANKUM-2M fermentor (Institute of Biological Instrumentation of RAS, Pushchino, Moscow region, Russia) with the initial volume of 6 L and the following cultivation parameters [11]: temperature 30 °C, concentration of dissolved oxygen of 50% saturation, pH 6.0, agitation rate of 800 rpm. During cultivation, pH was maintained automatically by the addition of 20% NaOH. Oxygen concentration was maintained by the regulation of air supply to the fermentor and the agitation rate.

### 2.3. Analytical Methods

Biomass was estimated by its dry weight [14]. The ammonium nitrogen was measured potentiometrically using an Orion ion-selective electrode (Thermo Fisher Scientific, Waltham, MA, USA) [14]. The concentration of glucose was measured by the enzymatic glucose oxidase method [14].

To determine citric acid (CA) and other acids, 6% HCLO_4_ was added to samples at a ratio of 1:1 and centrifuged at 6000× *g* to separate the precipitate. Quantitative determination of acids was carried out using an LKB chromatograph (LKB, Bromma, Sweden) with an Inertsil ODS-3 reverse phase column (250 × 4 mm) (Elsiko, Moscow, Russia). Mobile phase: 20 mM phosphoric acid; flow rate 1 mL/min; temperature 35 °C. Detection was carried out at 210 nm. CA, isocitric acid (ICA), and other acids were identified with the corresponding standards (reagents from Reakhim (Moscow, Russia) and Sigma-Aldrich (St. Louis, MO, USA)). Acid concentrations were calculated by calibration with external standards based on peak areas.

The cellular content of carbon, hydrogen, and nitrogen was determined using a C, H, and N analyzer purchased from Carlo Erba Strumentazione (Milan, Italy). The ash content was determined by burning the samples in a muffle furnace. The content of oxygen was calculated according to the equation:O = 100 − (C + H + N + ash),
where C, H, N are the contents of carbon, hydrogen, and nitrogen, respectively.

Methyl esters of fatty acids were obtained by the method of Sultanovich et al. [33]. Cells were vacuum-dried at 70 °C to constant weight and subjected to direct transesterification at 80 °C for 3 h in a reaction mixture (methanol:hydrochloric acid:chloroform (10:1:1, *v*/*v*/*v*)), which was supplemented with docosane (C_22_H_46_) as an internal standard. The resulting methyl esters of fatty acids were extracted three times from the reaction mixture with *n*-hexane, dried over sodium sulfate, evaporated on a vacuum evaporator, and analyzed by gas-liquid chromatography on a Chrom-5 gas-liquid chromatograph equipped with a flame ionization detector and a column (2 m × 3 mm) packed with a stationary phase representing 15% Reoplex 400 on Chromaton N-AW (0.16–0.200 mm). The column temperature was 200 °C. The identification of methyl esters of fatty acids was performed with the use of standard fatty acid mixtures from Serva (Heidelberg, Germany). The lipid content of the biomass was determined as the sum of fatty acids

For the amino acid assay, the biomass was freeze-dried. A total of 2 mL of 80% ethanol acidified with 0.1 N HCl was added to 20–30 mg of dry biomass and held for 24 h at room temperature. The extract was centrifuged, the residue was discarded, and the supernatant was assayed on a Biotronik LC2000 amino acid analyzer (Maintal, Germany) for free amino acids by the Spackman method. For the bound amino acid assay, the biomass residue after extraction of free amino acids with ethanol was additionally washed with 80% ethanol and dried at 65–70 °C. A total of 10–15 mg of the sample was hydrolyzed with 6 N HCl at 110 °C for 24 h. Excess HCl was removed from the extract using a rotary evaporator. A total of 2 mL of 0.2 N Na-citrate buffer (pH = 2.2) was added to the dry residue. The amino acid content was determined using a Biotronik LC2000 automatic amino acid analyzer (Maintal, Germany).

### 2.4. Calculations

The production parameters of CA biosynthesis, such as CA yield from glucose consumed (Y_CA/S_) (*g*/*g*), volume productivity (Qp) (g/L × h), and the specific rate of CA synthesis (qp) (g/g × h), were calculated according to the equations: Y_CA/S_ = p/s, Qp = p/(v × t), qp = p/(x × t), where p is the total amount of CA (g) in the culture liquid by the end of cultivation; s is the total amount of glucose consumed (g); x is the total amount of biomass (g) in the culture liquid by the end of cultivation; v is the initial volume of culture liquid (L); t is the cultivation time (h).

Specific growth rate (μ) was calculated as:(1)$$\mu =\frac{\ln(x_{2}-x_{1})\;}{t_{2}-t_{1}}$$
where x_2_ is the concentration of biomass (g/L) at any time, x_1_ is the concentration of biomass (g/L) at the initial time, and (t_2_ − t_1_) is the time (h) during which the biomass has increased.

The yield of biomass from the amount of glucose consumed (Y_X/S_) (*g/g*) was calculated as:Y_x/s_ = x/s,
where x is the total amount of biomass (g) in the culture liquid by the end of cultivation; s is the total amount of glucose consumed (g).

The energy capacity of biomass (Q_B_, kJ/g) was calculated by the formula [34]:Q_B_ = 0.28 × L + 15.1,
where L is the lipids in dry cell weight (DCW).

The energy yield of biomass (η_B_) (*g*/*g*) was calculated as:η_B_ = Q_B_/Q_S_ × Y_x/s_,
where Q_B_ is the energy capacity of biomass (kJ/g); Q_S_ is the energy capacity of glucose (kJ/g); Yx/s is the biomass yield from the amount of glucose consumed (*g*/*g*). The energy capacity of glucose (Q_S_) was taken to be 14.47 kJ/g [34].

### 2.5. Statistical Analysis

All data presented in this paper are the means of quadruplicate experiments ± standard deviation. One-way analysis of variance using the MS-Excel 2013 Data analysis tool was applied to evaluate the statistical significance of the variation pattern and the effect of nitrogen concentrations on CA production. The test was performed with the null hypothesis that no effect of changing the concentration of (NH_4_)_2_SO_4_ will be observed on CA production. The test confidence value was set to 95% with α = 0.05. The validation of the hypothesis was performed using the standard F-test for statistical variance.

## 3. Results

### 3.1. Effect of Nitrogen on Yeast Growth and Citric acid Production

The influence of nitrogen on the parameters of growth of *Y. lipolytica* VKM Y-2373 and citric acid (CA) production was investigated in a series of (NH_4_)_2_SO_4_ concentrations from 2 to 10 g/L. Figure 1 shows that the growth dynamics of this yeast strain did not depend on the nitrogen concentration within this range. Namely, exponential growth was observed from 3 to 12 h of cultivation; the phase of retarded growth lasted up to 24 h of cultivation (at 2–3 g/L (NH_4_)_2_SO_4_)) and to 36 h of cultivation (at the concentrations of (NH_4_)_2_SO_4_ above 4 g/L). At the end of the growth phase, nitrogen in the medium was exhausted, and all the glucose consumed was converted into cell biomass without the excretion of CA. In contrast, CA was actively excreted in the stationary growth phase (after 24 or 36 h in, depending on the nitrogen concentration used). The accumulation of CA was maximum at 144-th h of cultivation.

Table 1 summarizes data on the parameters of yeast growth and CA accumulation of *Y. lipolytica* VKM Y-2373. The biomass production was significantly different when the concentration of (NH_4_)_2_SO_4_ was increased from 2 to 10 g/L. Thus, the biomass production of cells grown at 10 g/L (NH_4_)_2_SO_4_ was 4.3 times higher than the cells grown at 2 g/L (NH_4_)_2_SO_4_.

The biomass yield (Y_X/S_) was higher by 143% at high concentrations of (NH_4_)_2_SO_4_ (6 and 10 g/L) than at low nitrogen concentrations. The maximum specific growth rate (μ_max_) in the phase of exponential growth weakly depended on the (NH_4_)_2_SO_4_ concentration and varied from 0.23 to 0.27 h^−1^.

Also, the accumulation of CA was significantly different in dependence on nitrogen. When the concentration of (NH_4_)_2_SO_4_ was increased from 2 to 4 g/L, the CA production elevated by 1.9 times. The subsequent increase in nitrogen concentration lightly repressed CA production (by 1.2 times). Thus, the optimal concentration of (NH_4_)_2_SO_4_ for CA accumulation by *Y. lipolytica* VKM Y-2373 was 4 g/L, with the maximum production of CA equal to 99.9 g/L.

As seen from Table 1, at 2–4 g/L (NH_4_)_2_SO_4_ in the medium, isocitric acid (ICA) concentration comprised 3.18–5.97 g/L. This concentration increased to 8.09 g/L at 10 g/L (NH_4_)_2_SO_4_.

As seen in Table 1, the maximum yield of CA from the glucose consumed (Y_CA/S_ = (0.72–0.78 *g*/*g*)) was observed at 3–6 g/L (NH_4_)_2_SO_4_ and decreased to 0.58 and 0.52 *g*/*g* at 2 and 10 g/L (NH_4_)_2_SO_4_. The specific rate of CA synthesis (q_p_) decreased by three times at 10 g/L (NH_4_)_2_SO_4_. The volume productivity of CA synthesis (Qp) was equal to (0.81–1.11) g/L × h at 3–6 g/L (NH_4_)_2_SO_4_ and decreased to 0.48 and 0.57 g/L × h at 2 and 10 g/L (NH_4_)_2_SO_4_, respectively.

One-way analysis of variance was applied to evaluate the statistical significance of the variation pattern and the effect of (NH_4_)_2_SO_4_ concentrations on CA production. The quadruplicate data for the net CA production was used for the statistical analysis. As seen in Table 2, the F ratio (of variances) value equal to 146.4481533 was higher than the F critical value equal to 3.055568276 as recommended by the Fisher’s table at the 95% confidence level and the dF(4,15) degree of freedom data set. In addition, the probability value (*p*-value) of the test was 0.00000000001, which was less than the set value of *p* (0.05) for this test. Therefore, the results indicate the null hypothesis can be easily rejected, and it can be concluded that the variation in (NH_4_)_2_SO_4_ concentrations has a significant effect on the CA production by *Y. lipolytica* VKM Y-2373.

### 3.2. Effect of Nitrogen Concentration on the Characteristics of Biomass of Producing Strain

Table 3 summarizes data on the relative content of protein, lipids, and macroelements in the biomass of *Y. lipolytica* VKM Y-2373 grown in media with different concentrations of (NH_4_)_2_SO_4_.

As seen in Table 3, the increase in the concentration of (NH_4_)_2_SO_4_ from 2 to 10 g/L weakly influenced the protein production of *Y. lipolytica* VKM Y-2373 (0.19–0.22 *g*/*g*) but greatly enhanced the total amount of protein in the cultivation medium (from 1.19 to 6.02 g/L). Similarly, the increase in the concentration of (NH_4_)_2_SO_4_ from 2 to 10 g/L did not influence the lipid production of *Y. lipolytica* VKM Y-2373 (0.1–0.11 *g*/*g*) but greatly enhanced the total amount of lipids in the cultivation medium (from 0.67 to 2.75 g/L). The energy capacity of biomass (Q_B_) is not varied (15.13 kJ/g) and is obviously correlated with lipid production. The energy yield of cells (_ηX/S_) increased from 0.07 to 0.18 *g*/*g* when the concentration of (NH_4_)_2_SO_4_ was changed from 2 to 10 g/L and obviously correlated with the dynamics of biomass yield (Y_X/S_) (compare relevant data in Table 1 and Table 3).

As seen in Table 3, the increase in the concentration of (NH_4_)_2_SO_4_ from 2 to 10 g/L led to the rise of the intracellular content of nitrogen from 3.08% to 5.08%; potassium from 1.44% to 1.93%; and magnesium from 0.11% to 0.20%. At the same time, the content of carbon (41.25–41.80%), hydrogen (5.93–6.20%), oxygen (40.25–41.25%), phosphorus (0.91–0.96%), and calcium (0.11–0.14%) was relatively stable.

Table 4 summarizes data on the effect of nitrogen on the fatty acid composition of biomass of the producing strain *Y. lipolytica* VKM Y-2373. At all nitrogen concentrations used, the major fatty acids are palmitic (C_16:0_), palmitoleic (C_16:1_), oleic (C_18:1_), and linoleic (C_18:2_). The increase in the concentration of (NH_4_)_2_SO_4_ from 2 to 10 g/L slightly decreased the percentage of palmitic (by 25%), palmitoleic (by 19%), and oleic (by 23%) acids, whereas the percentage of linoleic, linolenic, and stearic acids rose by 2.7, 34.6, and 10.5 times, respectively. The sum of unsaturated fatty acids and the conditional unsaturation index of lipids increased from 68.56% to 76.15% and from 2.76 to 3.79, respectively.

Results showed that the content of bound and free amino acids (AA) in the protein of *Y. lipolytica* VKM Y-2373 did not depend on the concentration of ammonium sulfate in the medium (data not shown). Therefore, in this article, we present only the results on the amino acid composition of *Y. lipolytica* VKM Y-2373 biomass under conditions of CA production with maximal volume productivity, i.e., at 6 g/L (NH_4_)_2_SO_4_.

As seen in Table 5, among the 18 amino acids detected, the major ones are glutamine (36.82 mg/g DCW), asparagine (25.95 mg/g DCW), and leucine (22.66 mg/g DCW). The content of valine, lysine, glycine, threonine, alanine, serine, phenylalanine, and isoleucine varied from 10 to 20 mg/g DCW. Histidine and methionine were found in the amount of 4–6 mg/g; arginine, cysteine, tryptophan, tyrosine, and proline were detected in trace amounts. The free AA comprised 5% of the total amount of AA. The total content of glutamine and alanine was 69%. It should be noted that the last two amino acids are synthesized by the direct amination of α-ketoglutarate and pyruvate, i.e., the metabolites of the oxidative pathway of glucose.

## 4. Discussion

This paper was devoted to the effect of nitrogen on the synthesis of citric acid (CA) (as well as proteins and lipids as the secondary products of cultivation) by the producing strain *Y. lipolytica* VKM Y-2373 grown in the medium with glucose as the carbon and energy source.

The growth of *Y. lipolytica* VKM Y-2373 was maximum at 10 g/L (NH_4_)_2_SO_4_ (Table 1). The specific growth rate attained the maximum (µ_max_ = 0.23–0.27 h^−1^) at the 12th h of cultivation independently of the content of nitrogen in the medium. The same results were obtained earlier for other strains of this species [7,11,14,32]. According to data available in the literature, high concentrations of glucose in the medium (200 g/L instead of 50 g/L) may diminish µ_max_ by 3.1 times [11]. The values of the biomass yield (Y_X/S_) equal 0.27–0.30 *g*/*g* for *Y. lipolytica* VKM Y-2373 corresponded to those of the other strains of this species grown on glucose [35,36]. A higher biomass yield (0.41 *g*/*g*) was reported for the natural strain *Y. lipolytica* H917 despite the low rate of glucose consumption [16].

CA was synthesized in the stationary growth phase when *Y. lipolytica* VKM Y-2373 was in nitrogen starvation (Figure 1), which agrees with literature data on the acid production from glucose [7,8,9,10,11,12,13,14,15,16,17] and other carbohydrates [22,23,24,25].

The concentration of (NH_4_)_2_SO_4_ in the medium affected CA synthesis, with the high concentration (10 g/L) stimulating yeast growth and decreasing acid production. The maximum accumulation of CA (95.5–99.9 g/L) by *Y. lipolytica* VKM Y-2373 with the yield Y_CA/S_ equal to 0.72–0.77 *g*/*g* was observed at 4–6 g/L (NH_4_)_2_SO_4_ (Table 1). It is one of the best results of CA biosynthesis in the medium with glucose as the carbon and energy source. For comparison, the natural strain *Y. lipolytica* H222 grown on glucose under optimal conditions accumulated 41 g/L CA with the yield Y_CA/S_ = 0.55 *g*/*g* [11]. Only during the long-term cultivation (553 h) in the fed-batch mode did the accumulation of CA reach 97 g/L [12]. Another natural strain *Y. lipolytica* W29 produced 49 g/L CA with the yield Y_CA/S_ = 0.85 *g*/*g* [35]. The active producer of CA is the recombinant strain *Y. lipolytica* PG86, which was characterized by superexpression of the pyruvate carboxylase gene and produced CA (95 g/L) with the yield Y_CA/S_ of 0.75 *g*/*g* [13]. Another transformant *Y. lipolytica* PR32 with superexpression of the pyruvate carboxylase gene produced CA (111.1 g/L), with a yield Y_CA/S_ of 0.93 *g*/*g* [37].

The concentration of (NH_4_)_2_SO_4_ in the medium also affected the synthesis of isocitric acid (ICA), generated as a byproduct. The content of ICAwas maximum at the high nitrogen concentration (Table 1), which agrees with literature data concerning other strains of *Y. lipolytica*. For example, the strain *Y. lipolytica* IMUFRJ 50682 showed the ratio CA:ICA = 1:11.5 at a high nitrogen concentration in the medium and CA:ICA = 1.2:1 at a low concentration [38]. The highest ratio CA:ICA = 49.6:1 has been reported for *C. lipolytica* DSM 3286 grown on glucose under deep nitrogen deficiency; this ratio increased with the time of cultivation [9].

The concentration of nitrogen had an impact on the content and composition of various intracellular components of *Y. lipolytica* VKM Y-2373. At 10 g/L (NH_4_)_2_SO_4_, the protein production in the biomass reaches 0.22 *g*/*g* DCW with the total amount of protein in the cultivation medium equal to 6.02 g/L (Table 3). The high amount of protein under the conditions of active CA production (81.98 g/L) by *Y. lipolytica* VKM Y-2373 is of great interest because it allows the practical use of waste biomass, for instance, in the food industry and agriculture. There is little information about the ability of citrate-producing strains of *Y. lipolytica* to synthesize protein in significant amounts. The highest protein production (up to 0.55 *g*/*g* DCW) has been reported for the marine strain *Y. lipolytica* SWJ-1b grown on glucose [39]. Suitable results were obtained in experiments on the protein synthesis by *Y. lipolytica* cultivated on the glucose-containing hydrolysates of rye and oats straw and bran (protein content consisted of 0.305–0.445 *g*/*g* DCW) [40], as well as during cultivation on glycerol-containing waste (up to 0.45 *g*/*g* DCW) [41].

The protein production of *Y. lipolytica* VKM Y-2373 was correlated with the intracellular content of nitrogen (Table 3). Data on the low content of intracellular nitrogen in the stationary-phase cells of *Y. lipolytica* VKM Y-2373 (3.08–5.08%) when the acid formation was maximum is consistent with the literature data. For example, the exponential-phase cells of *Candida oleophila* grown on glucose contained 7.45% of nitrogen, while 3.96% was in the phase of active acid formation [9]. Similarly, the exponential-phase cells of *Y. lipolytica* A-101 contained 8.1–8.3% of nitrogen, while 4.2–5.1% was in the phase of active acid formation [42].

The protein production of *Y. lipolytica* VKM Y-2373 was also correlated with the intracellular content of potassium and magnesium (Table 3). There is an opinion in the literature that the high intracellular content of these metal ions is necessary for the stability, correct folding, and proper functioning of RNA and protein molecules [43,44].

The low lipid production in *Y. lipolytica* VKM Y-2373 (0.10–0.11 *g*/*g*) (Table 3) does not contradict the data of Papanikolaou et al. [10] that the CA-producing strain *Y. lipolytica* ACA-DC 50109 cultivated at different concentrations of glucose (from 34 to 150 g/L) synthesized lipids in amounts from 0.05 to 0.12 *g*/*g*. On the other hand, Carsanba et al. [17] have reported a higher content of lipids in the CA-producing strains *Y. lipolytica* Zu110 and *Y. lipolytica* W29 grown on glucose (0.42 and 0.35 *g*/*g*, respectively).

Despite the low content of lipids in the biomass of *Y. lipolytica* VKM Y-2373 grown at 10 g/L (NH_4_)_2_SO_4_, the total amount of lipids in the medium was relatively high (2.75 g/L). In various CA-producing strains of *Y. lipolytica*, this value varied from 0.95 to 2.4 g/L [15,17]. The mathematical models for lipid production by *Y. lipolytica* based on the regulation of the N/C ratio are presented in the literature, and simulation results achieved productivity of 0.95 g/L × h [45]. It should be noted that other parameter—pH of the medium differentially affects CA and lipid production in *Y. lipolytica* W29, with CA production enhanced at a more neutral pH and lipid production enhanced at more acid pH [46].

Unsaturated fatty acids dominate in the lipids of *Y. lipolytica* VKM Y-2373 (Table 4). The index of unsaturation (the ratio of the sums of unsaturated and saturated fatty acids) for these lipids varied from 2.49 to 3.79 and corresponded to the respective indices of the CA-producing strain *Y. lipolytica* ACA-DC 50109 grown on glucose [10] and the protein-producing strain of *Y. lipolytica* grown on the rye and oats straw and bran hydrolysates [40]. The lipid fraction of *Y. lipolytica* VKM Y-2373 biomass grown at a high concentration of ammonium sulfate contained significant amounts of essential linoleic and linolenic fatty acids (Table 4). These polyunsaturated fatty acids are the most important in the human diet. They serve as energy sources, possess high physiological activity, maintain membrane fluidity, are cofactors of many enzymes, participate in the regulation of various enzymatic systems and permeability of cellular and mitochondrial membranes, and are involved in oxidative phosphorylation [47].

The protein fraction of *Y. lipolytica* VKM Y-2373 biomass is also unique and contains all essential amino acids (AA) in the required amounts (Table 5).

To estimate the biological value of *Y. lipolytica* VKM Y-2373 protein, its composition expressed as mg of amino acid per gram of protein was compared with that of ideal protein on the FAO scale and available literature data (Table 6).

The comparison of *Y. lipolytica* VKM Y-2373 protein with that of *Y. lipolytica* presented in the review by Jach et al. [48] shows that they are similar in the profile of essential amino acids. In addition, the content of essential amino acids in the protein of *Y. lipolytica* VKM Y-2373 is even higher than is required according to the requirements of FAO/WHO for human adults [49]: by three times in threonine; by two times in histidinelysine, isoleucine, leucine, lysine, valine, and the aromatic amino acids (AAA).

## 5. Conclusions

The novelty of the present work lies in the study of the effect of different concentrations of nitrogen on the wide range of parameters (CA biosynthesis, the production of biomass, the accumulation, and composition of proteins and lipids) in *Y. lipolytica* VKM Y-2373 grown in media with glucose. It showed for the first time that by choosing a certain limiting concentration of nitrogen in the medium, it is possible to develop a process for obtaining CA and biomass enriched with a high level of essential amino acids and unsaturated fatty acids. Such waste biomass can be used as a feed additive in agriculture, as well as in the food industry. Of great interest is the technological mode with 6 g/L (NH_4_)_2_SO_4_, which is characterized by high productivity (Qp) (1.11 g/L × h).

## Figures and Tables

**Figure 1 biomolecules-12-01421-f001:**
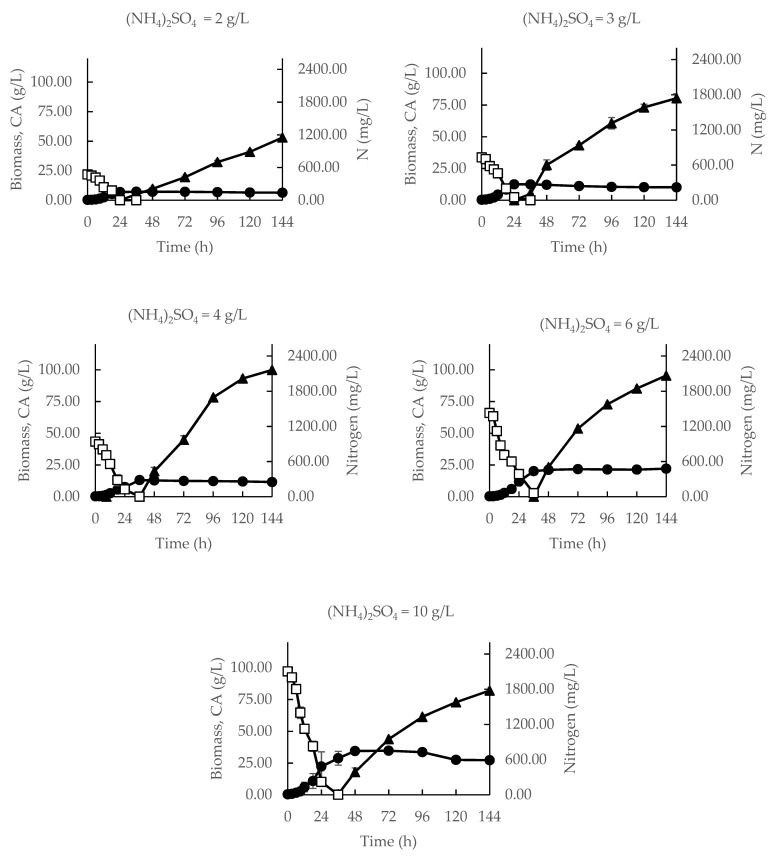
Citric acid production by yeast *Y. lipolytica* VKM Y-2373 at different concentrations of ammonium sulfate (●—biomass; ▲—CA; □—nitrogen).

**Table 1 biomolecules-12-01421-t001:** The effect of nitrogen concentration on growth- and CA production parameters.

(NH_4_)_2_SO_4_ (g/L)	Biomass (g/L)	CA (g/L)	µ_max_ (h^−1^)	Y_X/S_ (*g*/*g*)	ICA (g/L)	Y_CA/S_ (*g*/*g*)	q_p_ (g/g × h)	Q_p_ (g/L × h)
2	6.38 ± 0.32	53.05 ± 2.66	0.23 ± 0.01	0.07 ± 0.00	3.18 ± 0.15	0.58 ± 0.03	0.06 ± 0.00	0.48 ± 0.02
3	10.15 ± 0.10	80.30 ± 3.64	0.24 ± 0.02	0.10 ± 0.00	4.80 ± 0.23	0.78 ± 0.04	0.05 ± 0.00	0.81 ± 0.04
4	12.05 ± 1.27	99.90 ± 3.30	0.24 ± 0.02	0.10 ± 0.01	5.97 ± 0.24	0.77 ± 0.03	0.06 ± 0.00	0.86 ± 0.03
6	22.11 ± 1.73	95.48 ± 1.94	0.24 ± 0.02	0.17 ± 0.01	5.74 ± 0.11	0.72 ± 0.01	0.03 ± 0.00	1.11 ± 0.02
10	27.25 ± 0.74	81.98 ± 3.30	0.27 ± 0.02	0.17 ± 0.01	8.09 ± 0.38	0.52 ± 0.04	0.02 ± 0.00	0.57 ± 0.04

CA—citric acid; ICA—isocitric acid; µ_max_ —the maximum specific growth rate; Y_X/S_ —biomass yield; q_p_—specific rate of CA synthesis; Qp—volume productivity.

**Table 2 biomolecules-12-01421-t002:** One-way analysis of variance for CA production data.

**Summary**						
**Groups**	**Sample**	**SS**	**Avg.**	**Variance**		
2	4	212.2	53.05	7.0567		
3	4	321.2	80.3	13.2333		
4	4	399.6	99.9	10.8733		
6	4	381.9	95.475	3.7692		
10	4	327.9	81.975	10.9158		
**Analysis of Variance**						
**Source of Variance**	**Sum of Squares**	**dF**	**MSS**	**F**	** *p* **	**F Critical**
Between groups	5371.523	4	1342.88075	146.4481533	0.00000000001	3.055568276
Within groups	137.545	15	9.169666667			
Total	5509.068	19				

**Table 3 biomolecules-12-01421-t003:** The effect of nitrogen concentration on the composition of *Y. lipolytica* VKM Y-2373 biomass.

Parameters		(NH_4_)_2_SO_4_ (g/L)			
	2	3	4	6	10
Protein production in DCW (*g*/*g*)	0.19 ± 0.006	0.19 ± 0.006	0.19 ± 0.011	0.20 ± 0.004	0.22 ± 0.013
Lipid production in DCW (*g*/*g*)	0.11 ± 0.004	0.11 ± 0.003	0.11 ± 0.004	0.10 ± 0.004	0.10 ± 0.005
Total amount of protein (g/L)	1.19 ± 0.041	1.72 ± 0.071	2.26 ± 0.128	4.42 ± 0.094	6.02 ± 0.366
Total amount of lipids (g/L)	0.67 ± 0.022	0.99 ± 0.039	1.27 ± 0.043	2.29 ± 0.109	2.75 ± 0.133
Energy capacity of biomass (Q_B_) (kJ/g)	15.13 ± 0.01	15.13 ± 0.01	15.13 ± 0.01	15.13 ± 0.01	15.13 ± 0.01
η_X/S_ (*g*/*g*)	0.07	0.09	0.11	0.18	0.18
Carbon (% of DCW)	41.80 ± 0.48	41.79 ± 1.27	41.28 ± 2.83	41.30 ± 1.13	41.25 ± 2.90
Hydrogen (% of DCW)	6.20 ± 0.18	6.08 ± 0.34	5.95 ± 0.29	6.10 ± 0.28	5.93 ± 0.25
Nitrogen (% of DCW)	3.08 ± 0.17	3.28 ± 0.22	3.15 ± 0.07	4.15 ± 0.21	5.08 ± 0.25
Oxygen (% of DCW)	41.15 ± 0.59	41.25 ± 1.03	41.07 ± 2.83	40.25 ± 1.48	41.25 ± 2.90
Phosphorus (% of DCW)	0.91 ± 0.03	0.94 ± 0.02	0.96 ± 0.01	0.98 ± 0.01	0.96 ± 0.04
Potassium (% of DCW)	1.44 ± 0.06	1.48 ± 0.11	1.71 ± 0.06	1.70 ± 0.14	1.93 ± 0.11
Magnesium (% of DCW)	0.11 ± 0.01	0.13 ± 0.01	0.16 ± 0.01	0.19 ± 0.01	0.20 ± 0.01
Calcium (% of DCW)	0.11 ± 0.01	0.12 ± 0.01	0.10 ± 0.01	0.14 ± 0.01	0.12 ± 0.00

**Table 4 biomolecules-12-01421-t004:** Fatty acid composition of *Y. lipolytica* VKM Y-2373 at different nitrogen concentrations.

Fatty Acids (% from Total Fatty Acids)		(NH_4_)_2_SO_4_ (g/L)			
	2	3	4	6	10
Lauric acid (C_12:0_)	0.10 ± 0.00	0.13 ± 0.01	0.10 ± 0.00	0.12 ± 0.01	0.14 ± 0.01
Myristic acid (C_14:0_)	0.85 ± 0.07	0.75 ± 0.14	0.85 ± 0.07	0.93 ± 0.04	0.95 ± 0.02
Myristoleic acid (C_14:1_)	0.11 ± 0.01	0.11 ± 0.01	0.10 ± 0.00	0.25 ± 0.04	0.24 ± 0.02
Pentadecylic acid (C15:0)	0.11 ± 0.01	0.13 ± 0.01	0.10 ± 0.00	0.20 ± 0.01	0.18 ± 0.01
Palmitic acid (C_16:0_)	23.1 ± 0.99	24.4 ± 1.41	26.9 ± 0.42	21.55 ± 0.78	17.35 ± 1.48
Palmitoleic acid (C_16:1_)	12.9 ± 0.28	12.95 ± 0.04	13.75 ± 0.78	13.65 ± 0.49	10.41 ± 0.76
Heptadecanoic acid (C_17:0_)	0.58 ± 0.04	0.63 ± 0.03	0.58 ± 0.04	0.45 ± 0.07	0.43 ± 0.04
Heptadecenoic (C_17:1_)	1.20 ± 0.14	1.20 ± 0.14	1.05 ± 0.07	1.33 ± 0.11	1.55 ± 0.06
Stearic acid (C_18:0_)	0.11 ± 0.01	0.13 ± 0.01	0.14 ± 0.01	0.85 ± 0.07	1.15 ± 0.07
Oleic acid (C_18:1_)	45.45± 0.78	43.25± 1.20	44.35± 1.20	38.4 ± 1.27	33.55 ± 2.05
Linoleic acid (C_18:2_)	9.65 ± 0.21	10.9 ± 1.27	10.9 ± 1.26	17.15 ± 1.2	25.65 ± 0.92
Linolenic acid (C_18:3_)	0.11 ± 0.01	0.11 ± 0.00	0.31 ± 0.01	1.50 ± 0.07	3.80 ± 0.28
n-Nonadecanoic acid (C_19:1_)	1.15 ± 0.21	1.05 ± 0.07	0.95 ± 0.07	1.05 ± 0.07	0.93 ± 0.04
Arachidic acid (C_20:0_)	0.11 ± 0.01	0.12 ± 0.01	0.10 ± 0.00	0.21 ± 0.01	0.10 ± 0.00
Sum of unsaturated fatty acids (UFAs)	68.56	69.56	71.40	73.33	76.11
Sum of saturated fatty acids (SFAs)	24.85	26.15	28.67	24.18	20.15
Ration of UFAs/SFAs	2.76	2.67	2.49	3.03	3.79

**Table 5 biomolecules-12-01421-t005:** Amino acid composition of *Y. lipolytica* VKM Y-2373 at 6 g/L (NH_4_)_2_SO_4_.

Amino Acids (AA)	Bound AA	Free AA	Total Amount of AA
	mg/g DCW	mg/g DCW	mg/g DCW
Glutamine	31.43 ± 0.75	5.39 ± 0.62	36.82
Asparagine	25.69 ± 0.69	0.27 ± 0.08	25.95
Leucine	22.37 ± 0.66	0.30 ± 0.05	22.66
Valine	18.27 ± 0.23	1.14 ± 0.19	19.40
Lysine	18.39 ± 0.16	0.51 ± 0.10	18.90
Threonine	15.22 ± 0.34	0.34 ± 0.06	15.56
Glycine	15.02 ± 1.27	0.26 ± 0.08	15.28
Alanine	13.16 ± 0.37	1.53 ± 0.18	14.69
Serine	13.62 ± 0.30	0.30 ± 0.06	13.92
Phenylalanine	12.49 ± 0.16	0.10 ± 0.01	12.59
Isoleucine	12.09 ± 0.12	0.26 ± 0.01	12.34
Histidine	5.78 ± 0.08	0.22 ± 0.02	6.00
Methionine	4.36 ± 0.14	0.00	4.36
Arginine	0.11 ± 0.01	0.10 ± 0.00	0.21
Cysteine	0.10 ± 0.00	0.00	0.10
Tryptophan	0.10 ± 0.00	0.00	0.10
Tyrosine	0.10 ± 0.00	0.00	0.10
Proline	0.10 ± 0.00	0.00	0.10

**Table 6 biomolecules-12-01421-t006:** Comparison of essential amino acids profile of *Y. lipolytica* VKM Y-2373 with literature data and FAO standard.

Amino Acids (AA)	*Y. lipolytica* VKM Y-2373 (the Present Study) (mg/g Protein)	*Y. lipolytica* (mg/g Protein) [46]	AA Requirements for Adults from 2007 FAO/WHO (mg/g Protein) [47]
Arginine	1	48	n.d.
Histidine	27	26	15
Isoleucine	56	44	30
Leucine	103	68	59
Lysine	86	70	45
SAA	20	23	22
AAA	57	153	38
Threonine	71	48	23
Valine	88	53	39
Total	509	533	271

AAA—aromatic amino acids: tyrosine, phenylalanine, and tryptophan; FAO/WHO—Food and Agricultural Organization/World Health Organization; n.d.—not determined; SAA—sulfur amino acids: methionine and cysteine.

## Data Availability

Not applicable.

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
