# Peer review of "Effect of Nitrogen Concentration on the Biosynthesis of Citric Acid, Protein, and Lipids in the Yeast Yarrowia lipolytica"

_biomolecules, 2022, doi:10.3390/biom12101421_

Round 1
Reviewer 1 Report
The manuscript by Kamzolova and co-workers (Effect of Nitrogen Concentration on the Biosynthesis of Citric Acid, Protein, and Lipids in the Yeast Yarrowia lipolytica) is certainly within the scope of Biomolecules. In general the manuscript is well written, but there are several points that should be addressed before the manuscript is considered for publication.
Major Issues:
-The experimental setup and reported results are confusing. In lines 76-77 they claim that yeast cells were grown with 20 g/L glucose, however, in lines 87-88 they claim that “Glucose was added in portions (20 g/L) as it was consumed from the medium.”. However, in Figure 1 the authors did not show the dynamics of glucose consumption and the time-points were glucose was added (and its consumption). This is fundamental for the readers to understand how the cells could produce up to ~100 g/L of citric acid (obviously not from just 20 g/L glucose) during the stationary phase (thus in the absence of a nitrogen source). How many additions of glucose were done?, were the additions performed at the same time-points with the different nitrogen levels?. The glucose additions were completely consumed?. The authors need to show this!
-Another major criticism in the manuscript is that in lines 140-143 the authors claim that they performed a “Statistical analysis” to evaluate their results, but in their Results section none of their results (Tables 1 to 4) have any statistical analysis reported! The authors should identify results with significant differences with different letters, while those not significantly different with the same letters (within each parameter/compound), for example.
Minor Issues:
-In Figure 1, the first two panels (2 and 3 g/L ammonium sulfate) needs to be corrected, the triangles should be black, instead of white triangles.
-In Table 1 the values of μmax, Yx/s, ICA, Yca/s, qp and Qp should be given as means ± SD in order to support some of the claims made in the text (lines 166-185).
-In lines 198-199 (The energy capacity of biomass (QB) not varied (15.13 kJ/g) and obviously correlated with the lipid content.), I believe that is correlated with “Lipid in DCW”, and not with “lipid content” in the medium (see lines 134-135).
-In line 217 “Table 3. Cont.” should be removed.
-In line 224 “Table 4. Cont.” should be removed.
-In line 328 “Table 5. Cont.” should be removed.
-lines 248-249, 266-268, 333 (and many others) need revision.
-Finally, this reviewer missed two related publications (Modeling and optimization of lipid accumulation by Yarrowia lipolytica from glucose under nitrogen depletion conditions. Biotechnol Bioeng. 115:1137-1151, 2018; pH selectively regulates citric acid and lipid production in Yarrowia lipolytica W29 during nitrogen-limited growth on glucose. J Biotechnol. 290:10-15, 2019) that the authors should consider/comment in their manuscript (results/discussion).
Reviewer 2 Report
The authors in their original research article presented the effect of different concentrations of ammonium sulphate on the biomass growth, its amino acid composition, as well as, biosynthesis of lipids and citric acid in Yarrowia lipolytica VKM Y-2373. Provided results claimed well-known facts that the content of nitrogen results in the final amount of biomass and other metabolites, and nitrogen starvation led to citric acid accumulation. I would ask the authors to describe the novelty of the work due to the fact that the research problem was discussed in many other studies.
The aim of the work should be rephrased.
The methodology for lipids extraction was omitted in the manuscript. The same for the conditions for chromatography analysis.
Section 2.5. Statistical analysis – the authors claimed that the results were statistically analyzed and post-hoc test was applied then. Still, in the results section, there are no mentions of statistics, significance for the different results, etc.
I propose also to rephrase the conclusions part.
Round 2
Reviewer 1 Report
The authors addressed all suggestions, the manuscript can be accepted for publication.
Reviewer 2 Report
The authors corrected the manuscript according to the Reviewer's comments and suggestions. Unfortunately, in my humble opinion, the response and the corrections on the statistical analysis issues are insufficient. Currently, the section with statistical analysis reads as follows: “All data presented in this paper are the means of quadruplicate experiments ± standard deviation.” Moreover, in the response, the authors claimed that the results were statistically analyzed with a one-way ANOVA. If so, and the authors concluded that the nitrogen concentration affected the accumulation of biomass and metabolites, more in-depth statistical analysis should be performed. In the previous version of the manuscript, Tukey's test was mentioned. This method or another post hoc analysis should be performed to evaluate which results differed statistically significantly.
